# Hypertriglyceridemia and Atherosclerotic Carotid Artery Stenosis

**DOI:** 10.3390/ijms232416224

**Published:** 2022-12-19

**Authors:** Yoichi Miura, Hidenori Suzuki

**Affiliations:** Department of Neurosurgery, Mie University Graduate School of Medicine, 2-174 Edobashi, Tsu 514-8507, Mie, Japan

**Keywords:** hypertriglyceridemia, atherosclerosis, carotid artery stenosis

## Abstract

Both fasting and non-fasting hypertriglyceridemia have emerged as residual risk factors for atherosclerotic disease. However, it is unclear whether hypertriglyceridemia increases the risks of the progression of carotid artery stenosis. Statins are well known to prevent carotid plaque progression and improve carotid plaque instability. In addition, statin therapy is also known to reduce cerebrovascular events in patients with carotid artery stenosis and to improve clinical outcomes in patients undergoing revascularization procedures. On the other hand, there have been no randomized controlled trials showing that the combination of non-statin lipid-lowering drugs with statins has additional beneficial effects over statin monotherapy to prevent cerebrovascular events and stenosis progression in patients with carotid artery stenosis. In this article, the authors demonstrate the mechanisms of atherosclerosis formation associated with hypertriglyceridemia and the potential role of lipid-lowering drugs on carotid artery stenosis. The authors also review the articles reporting the relationships between hypertriglyceridemia and carotid artery stenosis.

## 1. Introduction

There are many interventions in the prevention and treatment of dyslipidemia. Particularly low-density lipoprotein cholesterol (LDL-C) is known as a well-established risk factor for cardiovascular and cerebrovascular diseases. Statins are the first-choice therapy to reduce the risks of atherosclerotic disease events by lowering LDL-C levels. However, even after achieving LDL-C levels below current guideline targets, atherosclerotic disease events have been not eradicated. Thus, elevation of serum triglyceride (TG) levels is now being accepted as a residual risk factor for atherosclerotic diseases. According to current guidelines, the optimal cut-off values to predict cardiovascular diseases are reported for a fasting serum TG value as 150 mg/dL and for a non-fasting serum TG value as 200 mg/dL by the American Heart Association (AHA) [1], and as 175 mg/dL by the European Atherosclerosis Society [2] and the Japan Atherosclerosis Society [3]. 

Atherosclerotic carotid artery stenosis (CAS) is a progressive narrowing of the carotid artery (CA) lumen due to atherosclerosis, characterized by the local thickening of an arterial wall. The lesion is frequently located at the carotid bifurcation or the internal CA. The association between atherosclerotic CAS and cerebrovascular events was first reported by Savory in 1856, followed by similar case reports, which re-emphasized the relationships between CAS and cerebrovascular events [4]. The prevalence of CAS greater than or equal to 50% is estimated to be 3.7–9% in the general population [3,5]. In the Framingham cohort study of 1116 subjects, the high prevalence of CAS was observed in relation to acute ischemic stroke (60%), coronary heart disease (18%), and atherosclerosis (11%) [5]. CAS progression can cause cerebrovascular events due to hemodynamic ischemia or plaque rupture. Moreover, CAS is reported to account for at least 15–20% of all ischemic strokes [6] and is promoted by hypertension, smoking, and diabetes mellitus (DM) [3,5,7]. In addition, certain hemodynamic features including turbulent blood flow or low wall shear stress also play key roles in CAS progression [8,9]. Furthermore, high levels of total cholesterol (TC) and LDL-C are well-known as risk factors for CAS progression, whereas the relationships between hypertriglyceridemia and CAS are not well understood. In this report, the authors discuss the mechanisms of atherosclerosis formation associated with hypertriglyceridemia and the potential role of lipid-lowering drugs on atherosclerotic CAS. In addition, the authors review the articles reporting the association of hypertriglyceridemia with atherosclerotic CAS. 

## 2. The Mechanisms of Atherosclerosis Formation Associated with TGs

The prevention and treatment of atherosclerotic CAS should be based on the knowledge of the pathogenesis of atherosclerosis formation. Atherosclerosis is a generalized disease characterized by the accumulation of lipids, fibrous elements, and calcification within the arterial wall. Therefore, various factors are involved in its pathogenesis. For all the mechanisms of atherosclerotic formation in CA that are similar to those in other arterial sites, the formation and development of atherosclerosis are rather complex processes [10,11,12]. Here, we focused on the mechanisms of atherosclerosis formation associated with TG and TG-rich lipoproteins (TRLs) (Figure 1).

TGs, esters derived from glycerol and three fatty acids, are an essential source of energy for utilization in skeletal muscle and stored in adipose tissues and muscle fibers. TGs, which are almost insoluble in plasma, are transported in lipoprotein particles in circulating blood. To date, accumulating evidence has shown that elevated circulating levels of TGs and TRLs are associated with increased risks of cardiovascular [13,14] and cerebrovascular diseases [15,16,17,18,19,20,21,22]. 

TRLs are secreted by intestinal enterocytes (chylomicrons (CMs)) and hepatocytes (very LDLs (VLDLs)). The core TGs comprising CMs and VLDLs are rapidly hydrolyzed by lipoprotein lipase (LPL) on the blood vessel wall. LPL, a key enzyme that drives TG hydrolysis, is synthesized in macrophages, adipocytes, and myocytes of adipose and muscle cells, and then transported to the endothelial cell surface. LPL transforms CM and VLDL particles into TRL remnants and smaller lipoprotein particles [23]. High TG levels reflect the presence of high levels of TRL remnant particles from CMs and VLDLs [24]. TRL remnants include more cholesterols and TGs and seem to be more proatherogenic than LDLs [10,25].

In physiological conditions, most TRL remnants are absorbed by the liver. However, when TRL remnants are retained in the circulating blood due to the overproduction and/or impaired clearance of TRL remnants, they accumulate in the subendothelial space, causing the formation of atherosclerotic plaques [26,27]. Although CMs (size = 75 to over 1000 nm) and larger VLDLs are generally thought to be too large to penetrate the arterial wall, their smaller TRL remnants are not only transported across the arterial wall but also promote the binding to and retention within the connective tissue matrix [28]. Endothelial transcytosis is generally restricted to lipoproteins smaller than approximately 70 nm in diameter [29]. Although the transcytic transport system has not previously attracted attention, it is now accepted that transcytosis plays a key role in the transendothelial transport of TRL remnants. The transcytic transport system can be divided into direct (caveolae-mediated transcytosis) and indirect (receptor-mediated transcytosis) pathways according to their transport routes, and TRL remnants are exocytosed on the opposite side of the endothelial cell layer [30,31]. In human femoral arteries with peripheral artery occlusive disease, the accumulation of TGs has been shown in atherosclerotic plaques [32]. TRL remnants, which contain approximately 40 times more cholesterol compared with LDLs [29], can penetrate the arterial wall and then be directly taken up by macrophages, followed by foam cell formation, without undergoing oxidative modification as observed in LDLs [33,34]. The accumulation of TRL remnant particles in atherosclerotic plaques has been proposed to play an important role in the inflammatory response, and may further link their structures to the development of atherosclerosis [35]. Thus, TGs and TRLs contribute directly to the formation and progression of atherosclerotic plaque.

On the other hand, TGs and TRLs also promote atherogenesis via indirect mechanisms. With the abundance of TRLs in the circulating blood, cholesterol ester transfer proteins exchange cholesterol esters in LDLs and high-density lipoprotein (HDL) particles for TGs [36]. As a result, the size and density of LDLs further decrease, turning into small dense LDLs (sd LDLs), whereas anti-atherogenic HDLs decrease [37]. sd LDLs are found in high-density LDL fractions, have enriched cholesteryl esters, a low binding affinity to LDL receptors, and continue to flow in the circulating blood for long periods of time. sd LDLs easily migrate into the subendothelial space to form atherosclerosis [18]. In addition, TRL lipolysis at the vascular luminal surface increases vascular endothelial permeability and causes a pro-inflammatory state in endothelial cells, manifested in an increased intracellular production of reactive oxygen species [37], the secretion of tumor necrosis factor-α, and the expression of cell adhesion molecules [38]. The elevation of TG levels also induces TRLs enriched with apolipoprotein (Apo) C-III. Apo C-III impairs lipolysis by LPL and the clearance of TRLs [39]. Furthermore, Apo C-III affects signaling pathways that lead to the activation of nuclear factor-κB and an increase in inflammatory molecules associated with the development of fatty streaks and promotion of atherosclerosis [40]. In addition, TRLs may suppress the athero-protective and anti-inflammatory effects of HDLs [41,42]. Elevated TG levels are also known to be associated with a procoagulant state by increasing the levels of tissue plasminogen activator inhibitors and factor VII, and activating factor VII phospholipid complexes [43], factor X [44], factor XII [45], and thrombin generation [46].

## 3. Impact of Elevated Levels of Non-Fasting Serum TG on Atherosclerosis

Serum TG levels gradually increase for three to six hours after a standard meal and then slowly return to their initial levels at six to eight hours after the meal [47]. Therefore, serum TG values used to be measured after eight to twelve hours of fasting in a clinical setting [2], and the treatment strategies for the prevention of atherosclerotic diseases were based on such measurements. However, recently, it has come to be known that the effect of diet on TG levels is less problematic in the absence of dyslipidemia and that non-fasting TG levels may be much higher than fasting TG levels only when in the presence of dyslipidemia or clearance impairment of lipid particles after a meal [2,36]. In addition, most of the day occurs in a non-fasting state for subjects who eat at least three meals a day, and, therefore, fasting TG levels may not reflect daily TG values. Epidemiological studies have also shown that elevated non-fasting TG levels are associated with an increased risk of cardiovascular disease [48,49,50]. 

The impact of elevated non-fasting serum TG levels on atherosclerosis can be explained by the following mechanisms. After dietary fat is hydrolyzed by pancreatic lipase, the products are absorbed into the cells lining the small intestine, where they are resynthesized into TGs. The TGs are incorporated into the cores of CMs, and the CMs are secreted into the mesenteric lymphatic system. Then, the CMs enter the circulating blood. CMs are Apo B-48-containing lipoproteins with a large TG core (from 80 to 95%) [51]. Apo B-48 is known to be the only specific marker in the nutritional investigation of CM metabolism [52] because apo B-48 is synthesized only in enterocytes. TG hydrolysis in CMs by LPL results in the production of CM remnants that are relatively rich in cholesteryl esters. Interestingly, it has been reported that VLDL and VLDL remnants also increase after a meal [23]. The mechanism could be explained by increased hepatic VLDL secretion as well as the increase in CM remnants, leading to the delayed catabolism of VLDL. Although approximately 80% of elevated serum TG levels after a meal are due to CMs [53], approximately 80% of increases in particle numbers are due to VLDL particles [54,55]. Increases in TRL remnants are accentuated in insulin-resistant states [56]. Then, the TRL remnant particles, which contain a significant amount of cholesteryl esters, can penetrate the arterial wall. Increased TRL remnants after a meal promote inflammatory reactions to increase the expression of interleukins-6 and -1β, intercellular adhesion molecule-1, monocyte chemoattractant protein-1, vascular cell adhesion molecule-1, and others [57], leading to the induction of apoptosis [58] or accentuation of the inflammatory response of endothelial cells to tumor necrosis factor-α [59]. Thus, the elevation of serum TG levels after a meal indicates an accumulation of proatherogenic TGL remnants (derived from CMs and VLDLs), as well as an increase in small dense LDLs [36] and a decrease in anti-atherogenic HDLs [37], causing proatherogenic processes including inflammatory reactions and endothelial dysfunction [60,61].

## 4. The Role of Lipid-Lowering Drugs in Patients with Atherosclerotic CAS

The treatment of atherosclerotic CAS should follow guidelines’ recommendations. The best medical treatment (BMT) including a statin, anti-platelet agent, and blood pressure-modifying agent is recommended for all patients with CAS to prevent cerebrovascular and cardiovascular events, regardless of whether these patients eventually undergo carotid endarterectomy (CEA) or CA stenting [62,63].

Statins not only have potent LDL-C-lowering properties but also significantly reduce TG levels. In addition, statins appear to beneficially influence fluctuating TG levels after meals. Chan et al. showed that atorvastatin significantly reduced plasma concentrations of apo B-48 in obese patients with dyslipidemia by accelerating the catabolism of CM remnants [64]. Similarly, Hogue et al. revealed that atorvastatin significantly reduced plasma apo B-48 levels in patients with type-2 DM because of a significant decrease in its production [65]. Thus, adjusted statin medication is consistently one of the main BMT components for patients with CAS [66,67]. 

Statins are well known not only to decrease carotid intima-media thickness (cIMT) and carotid plaque progression but also to improve carotid plaque instability [11]. Statin therapy is also reported to reduce cerebrovascular events in patients with CAS suffering a transient ischemic attack [68] and to improve clinical outcomes for patients with atherosclerotic CAS undergoing CEA or CA stenting [69]. These effects might be exerted by the pleiotropic effects such as a reduction in the inflammatory reaction, endothelial activation, leukocyte intra-plaque infiltration, and an increase in the protective migration of smooth muscle cells [70,71]. In the investigation of the effects of pravastatin on the composition of human carotid plaques removed by CEA in symptomatic patients, pravastatin was reported to decrease lipid content, lipid oxidation, the inflammatory reaction, matrix metalloproteinase*-2*, and cell death, and to increase the tissue inhibitor of metalloproteinase 1 and collagen content [70]. According to the guidelines of the European Atherosclerosis Society [2] and the Japan Atherosclerosis Society [3], statin therapy is appropriate for patients with atherosclerotic CAS who sustain an ischemic stroke to reduce LDL-C to a level less than or equal to 70 mg/dL, fasting TG to a level less than or equal to 150 mg/dL, and non-fasting TG to a level less than or equal to 175 mg/dL. 

In contrast, there is insufficient evidence to support the impact of non-statin lipid-lowering drugs on the prevention of CAS progression and cerebrovascular events due to CAS.

Omega-3 polyunsaturated fatty acids, eicosapentaenoic acids (EPAs), and docosahexaenoic acids are approved as adjuncts to diet for lowering serum TG values via multiple metabolic pathways [72]. A few studies reported that omega-3 fatty acids reduced non-fasting hypertriglyceridemia [73] and accelerated CM clearance [74]. Tinker et al. reported that postprandial decreases in apo B may be caused by the omega-3 fatty-acid-mediated inhibition of both hepatic and intestinal apo B secretion and synthesis [75]. Among omega-3 fatty acids, only EPA has been reported to reduce cardiovascular events [76,77], but its effect on CAS is unknown.

Fibrates, peroxisome proliferator-activated receptor (PPAR) alpha agonists, have been the most effective agents for lowering serum TG levels. Several studies revealed the efficacy of fibrates in reducing fasting and non-fasting serum TG levels by increasing LPL-mediated lipolysis, inhibiting apo C-III production, increasing hepatic beta-oxidation, and decreasing the secretion of VLDL particles [78]. Fibrates also increase serum HDL levels by approximately 6–10% [79,80]. However, it is unclear if fibrates reduce the risk of cerebrovascular events and prevent CAS progression. According to the VA-HIT study, fibrate therapy significantly reduced the risk of cerebrovascular events by 31% [80], while some meta-analyses reported that fibrate therapy had no significant effect on the incidence of stroke or fatal stroke [81,82]. Recently, the three-month results of an ongoing prospective single-arm study, consisting of 74 patients with a history of cerebrovascular diseases and fasting hypertriglyceridemia that was treated with pemafibrate, a selective agonist of PPAR alpha, were reported [83]. The study was conducted to investigate whether pemafibrate could prevent subsequent cerebrovascular events by reducing fasting serum TG levels: the baseline fasting TG levels were significantly higher in patients with intracranial artery stenosis (more than 227 mg/dL) but showed no difference in the prevalence of extracranial artery atherosclerosis and cerebral small vessel diseases. Pemafibrate significantly suppressed inflammatory markers, and the interim results were considered to be promising to prevent subsequent ischemic strokes [83]. More recently, the results of the phase-3 PROMINENT study were reported [84]. The 24-country, double-blind, randomized, and controlled trial enrolled 10497 type 2 DM patients who had mild to moderate hypertriglyceridemia and low serum levels of HDL-C and LDL-C. The patients were randomly assigned to a pemafibrate (5240 patients) or placebo (5257 patients) group for an average of 3.4 years. The results showed that pemafibrate reduced TG levels by 26.2% compared to the placebo. However, the primary endpoint, a composite of non-fatal myocardial infarction (MI), ischemic stroke, coronary revascularizations, and death due to cardiovascular events, occurred in 572 patients in the pemafibrate group and 560 patients in the control group, with no apparent effect modification in any prespecified subgroup. The overall incidence of serious adverse events was not significantly different between the groups, but pemafibrate did not appear to reduce the risk of cardiovascular disease in the trial [84].

Ezetimibe prevents intestinal cholesterol absorption, and ezetimibe alone or combined with a statin beneficially influences fasting and non-fasting hyperlipidemia [85,86]. However, several clinical trials demonstrated that the combination therapy of ezetimibe and a statin failed to show beneficial effects for the reduction of cIMT [87,88,89].

Niacin can reduce fasting and non-fasting TG levels by restricting the availability of free fatty acids for lipoprotein synthesis [90,91]. Villines et al. revealed that niacin therapy induced the regression of cIMT [92]. However, the beneficial effects on the reduction of cerebrovascular events remain uncertain. Furthermore, in a meta-analysis of 11 eligible trials including 9959 patients, niacin therapy was not associated with changes in the incidence of cerebrovascular events [93]. 

Thus, EPA, ezetimibe, fibrates, and niacin, which have been reported to improve dyslipidemia, do not provide sufficient evidence regarding the prevention of CAS progression as well as cerebrovascular events. In addition, to our knowledge, there have been no reports of other non-statin lipid-lowering drugs, such as cholesteryl ester transfer protein inhibitors and proprotein convertase subtilisin/kexin type 9, that showed a beneficial efficacy in preventing CAS progression and cerebrovascular events.

There are other drugs that are not commonly used for dyslipidemia but may reduce CAS progression. Cilostazol (CLZ), a selective inhibitor of phosphodiesterase 3, has an anti-platelet function and vasodilating effects. Recently, a meta-analysis of 5 randomized controlled trials with 698 patients revealed that CLZ can have beneficial effects in preventing cIMT progression and improving the lipid profile by decreasing serum TG, LDL-C, and TC levels [94]. Studies using animal models have also indicated that CLZ can reduce serum TG and TC levels [95] and inhibit neointimal formation [96]. In another report using a rat model, CLZ inhibited TG accumulation in the ligated CA without a reduction in serum TG levels [97].

## 5. Previous Reports Regarding the Association of Hypertriglyceridemia with Atherosclerotic CAS

cIMT is now being accepted as an independent risk factor for the development of cardiovascular and cerebrovascular diseases [98,99]. The meta-analyzed multiple-cohort study, USE-IMT, also reported that cIMT was a risk factor for MI and cerebrovascular events [100]. However, in the PROG-IMT study analyzing 21 eligible studies with 36984 patients during a mean follow-up period of 7 years, the association between cIMT progression and cardiovascular events remained unproven [101] (Table 1).

While contradictory results exist [102,103], it seems clear from the accumulating evidence to date that hypertriglyceridemia is a risk factor for ischemic stroke [104,105]. With regard to atherosclerotic CAS, recently, several studies have examined the association between serum TG levels and risks of CAS progression. A meta-analysis of 64 randomized controlled trials to investigate the relationship between baseline serum TG levels and cerebrovascular events found that each increase of 10 mg/dL in baseline serum TG levels was estimated to increase the risk of all strokes by 5% and to induce cIMT progression by 3.0 μm/year [15]. Teno et al. reported that cIMT in patients with elevated fasting serum TG levels was greater than that in patients with normal fasting serum TG levels in 61 patients with type 2 DM [16]. Vouillarmet et al. retrospectively analyzed 342 patients with DM with a 13.6-year follow-up period and revealed that CAS progression tended to be associated with higher fasting serum TG levels at the first duplex ultrasound [17]. Furthermore, Kitagami et al. retrospectively analyzed 71 patients with moderate to severe CAS with normal LDL-C levels and revealed that fasting serum TG levels greater than or equal to 150 mg/dL were an independent risk factor for CAS progression [18]. Thus, accumulated evidence has suggested that elevated serum fasting TG levels are associated with an increased risk of cerebrovascular events and CAS progression [15,16,17,18].

Recently, non-fasting as well as fasting serum TG levels have gained more and more attention as a residual risk factor for atherosclerotic diseases. Although hypertriglyceridemia was previously defined and diagnosed generally using only fasting TG levels greater than 150 mg/dL, present guidelines have described non-fasting TG levels of 175 or 200 mg/dL as the optimal cut-off points to predict cardiovascular events [1,2,3]. Moreover, the guidelines have recommended higher cut-off values for non-fasting TG than those for fasting TG for the screening and management of hypertriglyceridemia. In a prospective study of 11,068 Japanese subjects initially free of cardiovascular and cerebrovascular events for an average of 15.5 years, Iso et al. found that elevated non-fasting serum TG levels at medical examinations were independently associated with MI, angina pectoris, and sudden death, independent of TC levels [48]. 

On the other hand, to date, the relationship between non-fasting TG levels and CAS progression has not been well-established. Teno et al. reported that elevated non-fasting serum TG levels, despite normal fasting serum TG levels, were independently correlated with cIMT in 61 patients with type 2 DM [17]. After that, Mori et al. investigated the association between non-fasting remnant-like particle TG levels and cIMT in 68 patients with type 2 DM [19]. In the report, the enrolled patients were divided by premeal and 2-h postprandial TG levels into a normo-triglyceridemia group, a non-fasting hypertriglyceridemia group, and a fasting hypertriglyceridemia group, and the association between cIMT and hyperlipidemia was analyzed. As a result, it was found that cIMT thickness was significantly higher in both the fasting and non-fasting hypertriglyceridemia groups than in the normo-triglyceridemia group. The results suggested that remnants retention due to delayed TG metabolism appeared to be closely related to atherosclerosis and that non-fasting hyperlipidemia, as well as fasting hyperlipidemia, were considered an independent risk factor for the early development of atherosclerosis. Idei et al. reported the association between mean serum TG values during a 1 year period and cIMT by ultrasonography in 115 patients with type 2 DM [20]. They showed that the mean non-fasting TG values during a 1 year period were an independent risk factor for cIMT thickness and were superior to either fasting TG values or one-point non-fasting TG values for predicting the presence of carotid plaque in patients with type 2 DM [20]. Recently, the authors consecutively investigated 96 patients with normal LDL-C levels and atherosclerotic CAS of greater than or equal to 50% with cerebral angiography and found that non-fasting hypertriglyceridemia at the beginning of the follow-up was an independent risk factor for subsequent CAS progression [21]. Furthermore, the study reported that the cut-off values for non-fasting serum TG to discriminate CAS progression from no progression were 169.5 mg/dL for less severe-side CAS, with a baseline stenosis of under 50%, and 154.5 mg/dL for worse-side CAS, with a baseline stenosis of greater than or equal to 50%. The findings suggest that in advanced CAS, non-fasting serum TG levels should be more strictly controlled to prevent CAS progression [21]. More recently, the authors analyzed 111 CAs in 88 patients with atherosclerotic CAS with more than a one-year follow-up and revealed that the area [TG ≥ 175] of cumulative non-fasting serum TG values greater than 175 mg/dL during the follow-up period was an independent risk factor for CAS progression [20]. Interestingly, the area [TG ≥ 175] was superior to the mean of non-fasting serum TG values during the follow-up period [22]. Thus, the stable control of non-fasting TG levels below 175 mg/dL is considered crucial to prevent CAS progression.

Findings are accumulating to show that fasting and non-fasting hypertriglyceridemia are potential risk factors for CAS progression and cerebrovascular events associated with CAS. However, these lines of evidence are not well-established, and further investigation would be warranted in future prospective randomized trials, including for whether non-statin lipid-lowering therapy reduces the risk of CAS progression.

## 6. Conclusions

Hypertriglyceridemia has emerged as a residual risk factor for atherosclerotic disease. However, it is unclear whether TG-lowering therapy brings an additional benefit to statins for prevention against cerebrovascular events in patients with atherosclerotic CAS. Prospective larger studies are needed to clarify these issues.

## Figures and Tables

**Figure 1 ijms-23-16224-f001:**
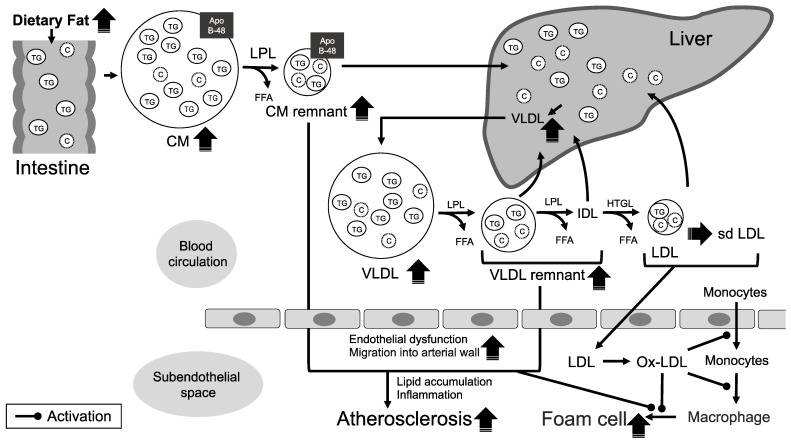
The mechanisms of atherosclerosis formation focusing on triglycerides (TGs). Dietary fats including TGs and cholesterol are absorbed into the cells of the small intestine and are transported in intestinally derived apolipoprotein (Apo) B48-containing chylomicrons (CMs). Then, CMs enter the circulating blood via the lymphatic system. On the other hand, very low-density lipoproteins (VLDLs) are synthesized by the liver and then secreted into the circulating blood. The core TGs comprising CMs and VLDLs are transformed by lipoprotein lipase (LPL) into smaller lipoprotein particles. After a meal, the major increases in lipoproteins occur not only for CMs and CM remnants, but also for VLDL and VLDL remnants. In healthy subjects, most of the TG-rich lipoprotein (TRL) remnants are absorbed by the liver. However, when TRL remnants are retained in the circulating blood due to the overproduction and/or impaired clearance of TRL remnants, TRL remnants can penetrate the arterial wall and accumulate in the subendothelial space. In addition, with the abundance of TRLs, the size and density of LDLs can further decrease because cholesterol ester transfer proteins exchange cholesterol esters in low-density lipoprotein (LDL) particles for TGs. Then, small dense LDLs (sd LDLs) easily migrate into the subendothelial space, where they are oxidized. The oxidized LDLs (ox-LDLs) promote the transendothelial migration of monocytes into the subendothelial space. Moreover, ox-LDLs induce the differentiation of monocytes into macrophages. Then, the accumulation of TRL remnant particles and sd LDL particles in the subendothelial space can contribute to the development of atherosclerosis. C, cholesterol; FFA, free fatty acid; HTGL, hepatic triglyceride lipase; and IDL, intermediate-density lipoprotein.

**Table 1 ijms-23-16224-t001:** Summary of previous reports regarding effects of hypertriglyceridemia on carotid atherosclerosis.

Study	Design	No. of Cases	Variables Associated with Carotid Atherosclerosis	OR or HR (95% CI)	*p* Value
Teno et al. [16]	Cross-sectional	61 patients with T2DM	Fasting TG ≥ 150 mg/dL	N/A	0.02
			Non-fasting TG ≥ 200 mg/dL	N/A	Tendency
Mori et al. [19]	Cross-sectional	68 patients with T2DM	Fasting TG ≥ 150 mg/dL	N/A	<0.05
			Non-fasting TG ≥ 200 mg/dL	N/A	<0.05
Idei et al. [20]	Retrospective	115 patients with T2DM	Mean non-fasting TG during follow-up	OR, 1.20 (1.05–1.37)	0.009
Vouillarmet et al. [17]	Retrospective	342 patients with T2DM	Fasting TG ≥ 150 mg/dL	N/A	Tendency
Kitagami et al. [18]	Retrospective	71 patients with CAS with LDL-C < 140 mg/dL	Fasting TG ≥ 150 mg/dL	HR, 6.228 (1.533−25.309)	0.011
Miura et al. [21]	Retrospective	121 carotid arteries in 96 patients with CAS	Non-fasting TG ≥ 175 mg/dL	OR, 4.703 (1.511−14.638)	0.008
Miura et al. [22]	Retrospective	111 carotid arteries in 88 patients with CAS	Area [TG ≥ 175] (cumulative non-fasting TG ≥ 175 mg/dL during follow-up) ≥ 6.35 year-mg/dL	OR, 4.21	0.003

Abbreviations: CAS, carotid artery stenosis; CI, confidence interval; HR, hazard ratio; LDL-C, low-density lipoprotein cholesterol; OR, odds ratio; T2DM, type 2 diabetes mellitus; and TG, triglyceride. N/A = not available.

## Data Availability

Not applicable.

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
