# Peer review of "Hypertriglyceridemia and Atherosclerotic Carotid Artery Stenosis"

_ijms, 2022, doi:10.3390/ijms232416224_

Round 1

Reviewer 1 Report

I would like to congratulate the authors of this manuscript for writing such a well-organized and comprehensive review article, covering most of the triglyceride contribution in atherosclerosis formation. 

The only comment I have is regarding Figure 1, where the abbreviations are embedded in the figure, which I think should be moved to the figure legends since the figure already looks complicated.

Author Response

Thank you so much for your suggestions. We have revised our paper in accordance with the suggestions offered, and the revisions are red-characterized.

  1. I would like to congratulate the authors of this manuscript for writing such a well-organized and comprehensive review article, covering most of the triglyceride contribution in atherosclerosis formation. The only comment I have is regarding Figure 1, where the abbreviations are embedded in the figure, which I think should be moved to the figure legends since the figure already looks complicated.

Answer:

According to your suggestions, we revised Figure 1 and the figure legends in pages 3 to 4.

Reviewer 2 Report

Some of my comments are inserted into the text.

The manuscript offers a lot of data taken from the literature, some terms are not used properly in the context, and the conclusions include 2 totally discordant phrases.

The article needs minor revisions regarding especially some phrases which needs to be reformulated.

Conclusions are too general and need to  be revised.

Author Response

Thank you so much for your suggestions. We have revised our paper in accordance with the suggestions offered, and the revisions are red-characterized.

  1. Some of my comments are inserted into the text.The manuscript offers a lot of data taken from the literature, some terms are not used properly in the context, and the conclusions include 2 totally discordant phrases. The article needs minor revisions regarding especially some phrases which needs to be reformulated. Conclusions are too general and need to be revised.

Answer:

According to your suggestions, we revised some phrases through the paper. We also revised the Conclusion section in page 8 as follows: “Hypertriglyceridemia has emerged as a residual risk factor for atherosclerotic disease. However, it is unclear whether TG-lowering therapy brings an additional benefit to statins for prevention against cerebrovascular events in patients with atherosclerotic CAS. Prospective larger studies are needed to clarify the issues.”